# Beliefs and Attitudes of Residents in Queensland, Australia, about Managing Dog and Cat Impacts on Native Wildlife

**DOI:** 10.3390/ani10091637

**Published:** 2020-09-11

**Authors:** Jennifer Carter, Mandy B. A. Paterson, John M. Morton, Francisco Gelves-Gomez

**Affiliations:** 1School of Social Sciences, University of the Sunshine Coast, Sippy Downs, QLD 4556, Australia; fgelvesg@usc.edu.au; 2Royal Society for the Prevention of Cruelty to Animals, Wacol, QLD 4073, Australia; mpaterson@rspcaqld.org.au; 3Jemora Pty Ltd., PO Box 2277, Geelong, VIC 3220, Australia; johnmorton.jemora@gmail.com

**Keywords:** domestic dogs, domestic cats, wildlife, management, attitudes, gender, age, responsible pet ownership

## Abstract

**Simple Summary:**

The acceptability of methods for managing cats’ and dogs’ undesired encounters with wildlife remains a contested issue. Despite a wealth of research on the effectiveness of management strategies, successful implementation is reliant on public perceptions and attitudes towards the different strategies. This paper reports on the results of a survey which sought to understand the attitudes of a self-selected group of residents in Queensland, Australia, towards various management actions for controlling dog and cat populations and behaviour (hereafter managing dogs and cats). Our respondents collectively grouped strategies into those that directly cause wild (i.e., feral) dog and cat deaths and those that allow wild dogs and cats to live a ‘natural’ life, with the acceptability of the first group of strategies varying by gender and age. These important variations in beliefs and attitudes require careful management within each community for the success of any program to control wild dogs or cats.

**Abstract:**

Many humans have created close relationships with wildlife and companion species. Notwithstanding that companion species were at some point themselves wild, some wild (i.e., feral) and domesticated (owned) dogs and cats now have significant impacts on wildlife. Many strategies exist to control the impact of dogs and cats on wildlife, but the successful implementation of management initiatives is tied to public opinions and the degree of acceptability of these measures. This paper reports the findings of a survey assessing the beliefs of residents in Queensland, Australia, about dog and cat impacts on wildlife, and their attitudes towards various strategies and options for controlling wild (i.e., feral) and domesticated (owned) dogs and cats. The responses of 590 participants were analysed. Our respondents collectively grouped strategies into those that directly cause wild dog and cat deaths and those that allow wild dogs and cats to live a ‘natural’ life, which is a variation on past research where respondents grouped strategies into lethal and non-lethal methods. Community acceptability of strategies that directly cause wild dog and cat deaths (each assessed using five-category Likert scores) was lower amongst females and respondents aged 34 years or less. Gender expectations in most places and cultures still predominately suggest that women are more ‘caring’, supportive of animal welfare, and perhaps cognizant that wild dogs and cats are also sentient creatures and appreciate the problematic tension between controlling wild and companion species. Age-related differences may reflect the changing social values of communities at different points in time. There was high support for regulations that enforce responsible pet ownership but not for the importance of pet-free suburbs, which the majority of respondents considered unimportant. These important variations in beliefs and attitudes require careful management within each community for the success of any program to control wild dogs or cats.

## 1. Introduction

Humans have created close relationships with dogs and cats. This closeness permeates human history and cultures, and thus, dogs and cats are perhaps amongst the most iconic companion animals to humans, dwelling in every part of the world where humans reside [1,2]. Along with their charisma, intrinsic value and other virtues, dogs and cats owned by humans can benefit humans in multiple ways, including better health, a sense of joy, and the ability to develop an affinity and a more profound relationship with other humans and living creatures [3,4,5]. While dogs and cats can live alongside humans as pampered pets, working animals, or as strays somewhat dependent on humans, they also can live independently to humans as wild (i.e., feral) animals.

In Australia, dogs and cats were introduced by humans during European colonisation [6,7], with Newsome [7], at that point, suggesting dogs as the same species as dingos given they can interbreed. Since then, dogs and cats have remained companion animals, but many have been released or escaped and now live in largely human-free environments, such as forested areas, scrub and bushland, agricultural areas, or even in and around urban areas. In particular, dogs have bred with the Australian native dog, the dingo, and the offspring are considered a hybrid animal, while many cats living in the wild are very large and aggressive in comparison with those who are companion animals. Dogs and cats in Australia can impact local wildlife (culturally understood in Australia as species that existed prior to European colonisation, but see [8]) through, for example, pathogen transmission [9,10], species behavioural changes [11,12], and, most commonly, predation and competition [1,13,14]. As a consequence of the numbers of free-roaming dogs and cats, their wide spatial distribution, and the potential consequences for biodiversity, these dogs and cats are considered key threats to wildlife in multiple locations [2,15], and unowned dogs and cats are considered pests by many governments (for example, in Queensland, under the *Queensland Biosecurity Act*). There is, therefore, a sense of urgency to address, manage, and reduce the adverse effects of dogs and cats on wildlife.

Whether dogs and cats are owned or unowned, they still have the ability to have a negative impact on wildlife. Many strategies exist to manage dog and cat populations, and to reduce their direct and indirect impacts on wildlife. Broadly grouped, management actions fit within two groups. Non-lethal methods aim to mostly modify cat and dog behaviour and reproductive capacities by restricting their ability to move, reproduce, and kill wildlife [16]. Amongst the most widely used management actions within this group are habitat modification, frightening devices, repellents, microchipping, containment, and fertility control through desexing [16,17,18,19,20]. These are used mostly with owned animals. For unowned dogs and cats, non-lethal methods include drug-induced fertility control [21] and Trap, Neuter, Release, known as TNR, programs [22,23,24]. This strategy is usually only used for cats, and involves trapping them, neutering them, and releasing them back to where they were trapped. It is favoured by people opposed to euthanasia. Lethal methods used for unowned dogs and cats seek the termination of cats’ and dogs’ lives through culling programs and actions that include poisoning or trapping, followed by humane killing [16]. While the efficacies of different management strategies have been measured, their effectiveness has only been measured against limited criteria. For instance, efficacy has been assessed as reductions in numbers of wildlife deaths because of predatorial behaviour, and as decreases in dog and cat population sizes (see, for example, [25,26,27]), yet some uncertainty remains over their relationship with various environmental factors (e.g., extreme weather conditions and natural disasters), practical constraints in deploying some of these management strategies, such as the willingness of people to cooperate, costs of various strategies, and time lags until beneficial effects occur [26].

Uncertainty about the effectiveness of these methods is usually due to complexities associated with their implementation, for instance, social, ecological, and practical factors that could limit their success. Importantly, these management strategies are often most effective when applied through an integrated management approach—a timely use of diverse strategies designed to integrate with the social and ecological complexities of the locations where management actions take place. Indeed, the management of dog and cat populations’ impacts on wildlife is not only a practical or technical endeavour. The key to the success of management initiatives remains tied to public opinions and the degree of acceptability towards these measures, as the “strategies can be dependent on the support of local communities” [28] (p. 180).

Contrary to the measured effectiveness of different management actions, the degree of acceptability of any of these actions by local communities remains a contentious and debated topic due to conflicting values [29]. In many instances, the reasons behind such contention are the volatile and changing nature of human beliefs and attitudes towards dogs, cats, and wildlife (and biodiversity in general). At the most general level, human beliefs and attitudes towards wildlife, and dog/cat impacts on wildlife and how to manage them, might be conditioned by cultural assumptions, such as worldviews [30], and the way in which people develop attitudes from their expectations, daily routines, and their particular set of values [31].

Past researchers have used cognitive hierarchy theory to frame their understanding of the acceptability of wildlife management strategies. This hierarchy suggests that values transcend context from which value orientations or belief patterns towards particular phenomena, such as pest species management, are formed [32]. These value orientations affect attitudes (evaluations of the phenomenon) and lead to social norms (shared beliefs), behavioural intentions, and behaviours. Estevez et al. [22] combined cognitive hierarchy theory with risk perception theory to argue that conflicts over invasive species management are largely derived from differing value systems and, to some extent, stakeholder perceptions of risk, noting that attitudes to risk are also influenced by cultural and personal experiences. Crowley et al. [33] noted conflicting values can change over time, depending on processes that trigger, polarise, or manage conflict. They outlined a conflict curve with stages of agreement or acceptance, expressed disagreement, positions formed, escalation or de-escalation, and the destructive stage of conflict between groups. They suggested conflict management around competing values occur prior to escalation, as well as to de-escalate conflict. Shackleton et al. [34] offered a conceptual framework that accounts for human value changes over time and space. In particular, they noted a need to focus on the individual, as individual values vary within groups, such as agriculturalists or wildlife protectionists, as much as between groups (noting also that groups’ and individuals’ values can co-construct each other). They also suggested understanding the species itself, the effects of the invasion, and the socio-cultural, landscape, and institutional/policy contexts to avoid and manage conflict and improve the effectiveness of strategies.

In the case of managing dog and cat populations’ impacts on wildlife, public opinions are conditioned by the strong bond that exists between humans and dogs and cats [2,14], public concerns for the welfare of both domestic animals and wildlife [35] and structural and cultural factors that affect people’s attitudes to (and uses of) different management strategies [30,36,37]. Animal welfare concerns, in particular, encompass multiple species and forms. There are public concerns that limiting the spatial ranges of dogs and cats effectively restricts some natural behaviours. There are also public concerns over the suffering of animals when they are poisoned, and the welfare of native wildlife, including suffering when preyed upon or when they experience other adverse effects, and conflicts between values-based positions. A close relationship with pets can increase concern towards other animals or can be contested as a practice that causes prey death, demonstrating often-polarised values.

Drivers of people’s beliefs and attitudes towards wildlife, cats and dogs, and potential management actions have been researched. There are differences in community opinions about the appropriateness of management actions depending on whether dogs and cats are defined as feral or not [14,23,24,38], with even the definition of ‘feral’ contested, and no clear agreement on interpretation of the term ‘feral’ as it might be used differently in different countries [26]. Depending on the context, the term has been used to refer both to animals from domesticated species who themselves are no longer domestic and instead are living in a wild, free-roaming state (and often seen as a ‘pest’) and to domestic animals who are not contained and wander the vicinities of their primary residence but are ‘owned’ by a human [14,39].

Public perceptions about cat management strategies vary by region [14,35]. Research on cat management in North America has mostly focused on feral and free-ranging cats, and perceptions of people about controlling cat populations via strategies such as TNR [35,40,41]. Walker et al. [42] found that an individual’s support for various cat management strategies can be apparently discordant. For instance, a person may support mandatory desexing but not support measures such as domestic cat confinement. Those same researchers also found lower acceptability towards TNR than was anticipated and argued that the way TNR has been presented has the potential to affect people’s perceptions of that strategy. In Australia, research efforts have been dedicated to understanding the behaviours of dogs and cats as predators of native wildlife, and social research has assessed the views of the public about the need to address the impacts that dogs and cats have on wildlife. Hall et al.’s [43] (p.23) research into attitudes towards cat predation on wildlife found that the majority of Australian pet carers and non-carers “are more accepting of measures to restrict cats in the interests of wildlife protection” than pet carers and non-carers living in New Zealand, the UK, or the USA. Other research describes a divergence of attitudes between different groups of people (for instance animal activists and other members of the community), as well as differences in attitudes when no benefits to pets are perceived [44,45]. The degree of acceptability of different management strategies has also been shown to vary with gender [14,46] and according to whether or not people are carers of dogs and cats [47].

In addition to the many drivers shaping peoples’ beliefs and attitudes about dog and cat management interventions, there are incongruences between community members’ views and experts’ opinions [48], and also between the imperative to seek measures that recognise people’s concerns while improving public legitimacy and thus the effectiveness of management strategies [33]. There is a need to further assess potential determinants of people’s attitudes towards dog and cat management actions and various strategies for reducing or preventing dog and cat populations’ impacts on wildlife. These underlying attitudes may well not be explicitly recognised even by the person whose beliefs and attitudes are being assessed. As such, they can be considered ‘latent’. Understanding any important latent variables is important. Our prior view was that a person’s degree of acceptance of various possible strategies is determined, in part, by their underlying value orientations or beliefs which affect their degree of acceptance of strategies that cause wild dog and cat deaths. In the same way, it is possible that a person’s view of the importance of various options for controlling domestic (owned) dogs and cats is driven by the person’s underlying beliefs and consequent attitudes. As others have argued (see, for example, [14,36]), we also contend that managing dog/cat encounters with wildlife is not only a matter of managing the animals, but also about addressing the human behaviours that are the root of the problem. Thus, there is a need to understand socio-demographic factors that shape human attitudes towards dog and cat management strategies. The aims of the current study were: (a) to describes the attitudes of a self-selected group of adult residents in Queensland, Australia, towards dog and cat management strategies; (b) to assess whether there are underlying patterns in beliefs and attitudes of respondents about dog and cat management strategies, when these are assessed collectively, that may reflect underlying latent determining variables; and (c) to assess whether socio-demographic factors are determinants of some of these beliefs and attitudes.

## 2. Materials and Methods

### 2.1. Study Overview

The attitudes of a self-selected group of adult residents of Queensland, Australia, in late 2015/early 2016 were explored using an on-line questionnaire administered using Survey Monkey^®^ (San Mateo, CA, USA). The questionnaire was developed based on previous research discussed above with the questions arranged in four major sections as follows:The respondents’ views about general risk of extinction of wildlife and their degree of concern about this (3 questions).The risks to wildlife of various anthropogenic and natural occurrences (7 questions).The acceptability of various possible strategies for reducing or preventing wild dog and cat predation on wildlife (7 questions).The importance of various options for controlling domesticated (owned) dogs and cats (11 questions).

The questionnaire also requested details about participant demographics (gender, age, postcode) and their dog and cat ownership status, as well as whether they currently fed a cat they do not own. Finally, there were questions assessing the Royal Society for Prevention of Cruelty to Animals, Queensland’s (RSPCA Qld) Living with Wildlife program activities that could only be answered by the subset of participants who were aware of the program (results not reported in this paper).

The questionnaire consisted of multiple choice questions (where respondents ticked one or more options from a list), and 4- and 5-category Likert scales (where respondents could select one option to indicate their view) as these were deemed most suitable for the context and opinions being measured [49]. Regardless of the selected multiple choices or Likert scale choice, the ‘other’ option was also offered for most questions, where respondents were free to write whatever they wanted, and there were seven open questions.

Once developed, the questionnaire was tested with a number of RSPCA Qld staff members and veterinary students to ensure that it was easy to understand and answer. Some modifications were made to a few questions to improve clarity and remove any ambiguity following this pilot testing.

The research was carried out under the auspices of RSPCA Qld and adhered to the National Statement on Ethical Conduct of Research (2007–2015, and the update 2007–2018), and also the Australian Code for the Responsible Conduct of Research (2018). The email invitation included information on the research. Participation was completely voluntary.

### 2.2. Enrolment of Participants

We attempted to recruit participants from the general public rather than from known animal lovers and followers of RSPCA Qld. A snowball sampling technique was adopted. The URL for the questionnaire, along with a brief explanation of the research, was initially sent to contacts of the second-named researcher (M.P.) who were known to not be dog or cat owners, and who were known by M.P. through other social settings not attached to the RSPCA, such as sport and dance activities. These people were asked to forward it to their contacts. It was also distributed on social media. The questionnaire was available for participants from October 2015 until February 2016. The questionnaire preamble stated that only people aged 18 years or above were eligible to respond.

The survey was identified as coming from the RSPCA; this was included to add credibility and encourage people to participate as the RSPCA is a well-respected brand. It was recognised that this may result in some degree of selection bias, but as the questionnaire did not ask for respondents to identify themselves in any way and complete anonymity was ensured, it was felt that the benefits of the brand recognition on recruitment outweighed any downsides. The RSPCA Qld has a mandate of welfare for both wildlife and companion animals, so it was considered that both those concerned about damage to wildlife and those with high regard for companion animals would respond, and regard for both would be seen as desirable, reducing desirability bias.

### 2.3. Statistical Methods

Responses to all questions were coded numerically, e.g., age categories (determined with respect to Australian Bureau of Statistics categories) of 21 years and under were coded as 1, 22–34 years given 2 and so forth. Each response category (including undecided) on the Likert scales was given a number, and non-responses were set as missing values. Statistical analyses were performed using Stata (version 15; StataCorp, TX, USA). Correlations between variables were assessed using Spearman’s correlation coefficients, calculated using Stata’s -spearman and -ci2- commands. We used a principal component analysis (PCA) to assess whether there are underlying patterns in beliefs and attitudes of respondents about dog and cat management strategies that may reflect underlying latent determining variables with Stata’s -pca- command. PCA involves analysing responses to multiple questions collectively. It requires variables that are continuous data but our acceptability and importance variables were ordinal; however, fitting binary data are a reasonable alternative with PCA [50] and we first collapsed our variables to binary data reflecting acceptance (strongly acceptable/acceptable) versus otherwise (strongly unacceptable/unacceptable/undecided) and reflecting importance (very important/important) versus otherwise (unimportant/undecided). We also used a factor analysis for the same purpose with the original ordinal variables. Stata’s -factor- command was used with uniqueness estimated from the squared multiple correlations. As females and dog and cat owners were overrepresented in the study population relative to the Queensland population in 2016 (Table 1), and as a substantial number of respondents were females who owned dogs and/or cats, these analyses were also performed limited to females who owned dogs and/or cats.

Composite outcome (i.e., dependent) variables for risk factor analyses were defined based jointly on variability in responses, correlations, and PCA results. Associations between each socio-demographic independent variable (respondent’s gender, age, dog and cat ownership status, and their suburb’s socioeconomic status) and these outcome variables were assessed using multivariable multinomial (polytomous) logistic regression models, fitted using Stata’s -mlogit- command. Multinomial logistic regression was used as the outcome variables were categorical with more than two categories, and there was no inherent order or ranking across all categories of those variables. For each model, all five socio-demographic independent variables of interest were fitted simultaneously as there was no a priori reason to postulate that the effects of some of these independent variables on our dependent variables were mediated in part by their effects on the other independent variables (making the inferred directed acyclic graphs [51,52] straightforward), and because there was minimal collinearity between independent variables. The calibration (goodness of fit) of each model was assessed using a generalised Hosmer–Lemeshow goodness-of-fit test for multinomial logistic regression models [53], performed with Stata’s -mlogitgof- command. Deciles of the complement of the estimated probabilities of the reference outcome were used to generate the table to assess the fit. Pearson’s chi-squared statistic was used to compare observed and estimated frequencies in that table.

## 3. Results

### 3.1. Description of Respondents

Eight hundred and sixty-eight responses were received. Of the questions directed to all participants, 31 were considered highest priority (all 28 questions in the four major sections listed above, plus dog ownership, cat ownership, and whether they currently fed a cat that they did not own), and the 53 respondents who replied to less than 29 of these questions were excluded. Of the remaining respondents, those whose suburb postcode was not a Queensland location (*n* = 151) or did not supply a postcode (*n* = 72) were excluded to avoid any influence of state-based legislation, leaving 590 respondents used for analyses.

Demographics and dog and cat ownership status for these respondents are summarised in Table 1. The majority of the respondents were female (77%), and most were dog and/or cat owners with only 22% owning neither dogs nor cats. A wide range of ages were represented, and the socio-economic status of respondents’ postcodes varied widely. Relative to the Queensland population in 2016, females were over-represented in the study population, as were dog and cat owners (Table 1).

### 3.2. Risk of Extinction of Wildlife

Of the 589 responses to the question “In how much danger of extinction do you think our wildlife is in?” (Question 1), 49% of respondents (287) selected “great danger” and 49% (287) selected “some danger”, with 9 selecting “no danger” and 6 selecting “undecided”.

Of the 588 responses to the question “How concerned are you with the level of danger wildlife are in?” (Question 2), 61% of respondents (359) selected “very (concerned)” and 29% (170) selected “somewhat (concerned)”, with 9% (50) selecting “a little (concerned)” and 1.5% (9) selecting “not concerned”.

Responses to this second question were also assessed by response to the first question. Of the 287 respondents that selected great danger of extinction for the first question, 89% (256) were very concerned, while of the 287 respondents that selected some danger of extinction, 36% (102) were very concerned and 48% (138) were somewhat concerned.

### 3.3. Risks to Wildlife of Various Anthropogenic and Natural Occurrences

Respondents’ views about the impacts of various anthropogenic and natural occurrences on wildlife are summarised in Table 2. Substantial or high proportions of respondents considered that human encroachment and development in native habitats and predation by cats and dogs were having great impacts on wildlife, much higher than for the other possible sources of impact that we listed. For all possible sources of impact except global warming, however, almost all respondents considered that there is at least some impact. For global warming, 8% of respondents were undecided and 9% considered that it has no impact on wildlife (Table 2).

After excluding undecided responses, responses were not closely correlated between possible sources of impact; the closest correlation (i.e., the correlation where the absolute value of the correlation coefficient estimate was nearest to 1) was between diseases including introduced diseases and pollution (Spearman’s correlation coefficient: 0.48; 95% CI 0.42 to 0.54).

### 3.4. Acceptability of Various Strategies for Reducing or Preventing Wild Dog and Cat Predation on Wildlife

Respondents’ views about the acceptability of various strategies for reducing or preventing wild dog and cat predation on wildlife are shown in Table 3. Respondents were asked: “However you rated the causes above, predation by dogs and cats plays a role in killing and maiming wild animals. Please indicate how acceptable the following control measures are for wild dogs and cats.”. For all strategies, relatively few respondents were undecided. For trapping followed by humane killing and reproductive control, the majority of responses selected acceptable or strongly acceptable, while for poisoning, introducing lethal disease, and introducing a new predator, the majority of responses selected unacceptable or strongly unacceptable. In contrast, responses were more evenly spread across the 4 acceptability categories for the strategies ‘shooting’ and ‘trapping followed by desexing and returning to the wild’.

The correlation coefficients for correlations in responses between strategies are shown in Table 4. Responses were not closely correlated, with the closest correlations between shooting, trapping followed by humane killing, poisoning, and introducing lethal disease (Spearman correlation coefficients 0.43 to 0.72). For all other pair-wise comparisons, absolute values of Spearman correlation coefficients ranged from 0.10 to 0.39.

From the PCA, the first two components accounted for 37% and 16%, respectively, of the total variance of 7 (i.e., the sum of the variances of the 7 variables, each standardised to have a mean of 0 and variance of 1); eigenvalues for the first two components were 2.6 and 1.1, respectively. Loadings for these 2 components are shown in Table 5. The principal component scores for each respondent were calculated as the weighted sums of the respondent’s responses for the 7 strategies (standardised), weighted by the respective loading for each. Thus, strategies with loadings furthest from 0 (either positive or negative) have most effect on the respondents’ principal component scores, and strategies with loadings close to 0 have minimal effect on the principal component scores. Loadings can vary from −1 to 1 and, for any particular component, the loadings squared sum to 1, so a variable with a loading of, for example, 0.9 (or −0.9) would have a very large influence on the principal component scores for respondents for that component. The first component had moderately high positive loadings (0.37 to 0.50) for shooting, trapping followed by humane killing, poisoning and introducing lethal disease, and that component can be interpreted as the degree of acceptance of strategies that cause dog and cat deaths directly due to human intervention. The second component had moderately high positive loadings (0.54 to 0.61) for reproductive control (such as immunocontraception), trapping followed by desexing and returning to the wild, and introducing a new predator. That component can be interpreted as the degree of acceptance of strategies that allow wild dogs and cats to live a ‘natural’ life (including being preyed upon). Each principal component was calculated such that it was not correlated with any other component. Accordingly, this second component represents an additional underlying (i.e., latent) variable over and above, and independent of, that represented by the first component. Thus, respondents’ locations along the latent variable continuum that reflected their degree of acceptance of strategies that cause dog and cat deaths directly due to human intervention were independent of their locations along the latent variable that reflected their degree of acceptance of strategies that allow the animal to live a ‘natural’ life. Compared with respondents with low values for the first latent variable, those with high values for that variable were no more likely to have high values for the second latent variable. Loadings from the factor analysis reflected the same latent variables but loading patterns for factor 2 were less marked. Also from the factor analysis, uniqueness values were low to moderate for strategies that cause deaths directly (0.36 to 0.64) but were high for the three strategies that allow the dog or cat to live a ‘natural’ life (0.76 to 0.91), indicating that factors 1 and 2 did not jointly explain the variance well for any of those three strategies. PCA and factor analysis were repeated within females who owned dogs and/or cats (*n* = 331). Results were similar (mostly very similar) to those when all 527 eligible respondents were used as reported above and in Table 5 (results not detailed).

Based on these findings, a four-category outcome (i.e., dependent) variable was generated to capture both components. Each respondent was classified as having high (at least 3 strongly acceptable or acceptable) or low (less than 3) acceptance of the four strategies—shooting, trapping followed by humane killing, poisoning and introducing lethal disease—and high (at least 2) or low (0 or 1) acceptance of the three strategies—reproductive control, trapping followed by desexing and returning to the wild, and introducing a new predator. Distributions of this response (i.e., dependent) variable were compared by five independent variables: respondent’s gender, age, dog and cat ownership status, and their suburb’s socioeconomic status. Only respondents that provided responses to all 7 strategies were used for these analyses (*n* = 527). The results of the analyses are shown in Table 6. The model had good fit with observed frequencies similar to expected frequencies for most of the 40 decile-dependent variable combinations. P for testing the null hypothesis that observed frequencies equal expected frequencies (i.e., that the model was correct) was 0.154. Table 6 shows relative risk ratios. For example, the crude relative risk ratio was 0.3 for females being ‘high’/‘low’ rather than ‘low’/‘low’ (Table 6) and was calculated as the ratio of the relative risk of ‘high’/’low’ responses for females relative to males (15%/30%) to the relative risk of ‘low’/‘low’ responses for females relative to males (35%/22%). After adjustment for the other four independent variables, the adjusted relative risk ratio estimate was 0.4. Based on this adjusted estimate, the associated 95% CI and *p*-value, and prior evidence, this showed that, relative to males, females are less likely to have high acceptability for strategies that cause dog and cat deaths directly due to human intervention. Relative to respondents aged 34 years and under, those aged 55 to 64 years were more likely to have high acceptability for strategies that cause dog and cat deaths directly due to human intervention (relative risk ratio estimates 3.2 and 3.0, respectively). Relative to those who did not own cats, cat owners were less likely to have high acceptability for strategies that cause dog and cat deaths directly due to human intervention (relative risk ratio estimates 0.4 and 0.2, respectively).

In contrast, these results provided no evidence that respondents’ gender, age, dog and cat ownership status, and their suburb’s socioeconomic status are associated with their degree of acceptance of strategies that do not cause dog and cat deaths directly due to human intervention. For example, compared to males, females had a relative risk ratio estimate of 1.0 (95% CI 0.6 to 1.8) for high (rather than low) acceptability of strategies that do not directly cause death (i.e., for selecting ‘low’/‘high’ rather than ‘low’/‘low’). If the relative risk is known to be 1, the risk of high acceptability would be the same for females as males. This is in contrast with strategies that cause death where females were less likely than males to have high acceptability (as indicated by relative risks markedly less than 1 and low associated *p*-values for each of ‘high’/‘low’ and ‘high/‘high’, both rather than ‘low’/‘low’). Similarly, relative to respondents aged 34 years and under, there was no evidence that those aged 55 to 64 years were more or less likely to have high acceptability for strategies that do not cause dog and cat deaths directly (relative risk estimate 1.5; 95% CI 0.8 to 2.8), which is in contrast with strategies that cause dog and cat deaths directly where relative risk estimates for those aged 55 to 64 years were well above 1 and associated *p*-values were low (Table 6). However, as demonstrated by the 95% CIs, these results do not preclude the possibility of the true associations actually being modestly strong.

### 3.5. Importance of Various Options for Controlling Domestic (Owned) Dog and Cats

Respondents’ views about the importance of various options for controlling domestic (owned) dog and cats are shown in Table 7. In summary, respondents were highly supportive of options that ensure responsible pet ownership. Respondents were asked: “Now thinking about pet dogs and cats, please indicate how important you think the following possible control measures are.”. Confinement solutions were considered important or very important by most respondents: confining dogs to the owner’s property (96% of respondents), confining cats to the owners’ property (94%), confining cats to the home (94%), and confining cats within an enclosure (83%). Other than this latter option, disregarding the distinction between ‘important’ and ‘very important’, the only dog- and cat-specific options with important variation were compulsory registration of dogs and cats by local council and ‘cats required to wear a bell at all times’. Within respondents, opinions about compulsory registration were generally consistent for dogs and cats, but 8% (46) of the 585 respondents that provided responses to both options considered that compulsory registration was important or very important for dogs but was unimportant or they were undecided for cats (Table 8).

There was more variation in views about options that we offered that would apply to both species (Table 7). The majority of respondents considered development of suburbs/areas that are completely pet free unimportant.

Associations between socio-demographic variables (independent variables) and respondents’ collective views about the importance of compulsory registration of dogs and cats by local councils (the response or dependent variable) are shown in Table 9. Of the 590 study respondents, 6 were excluded from these analyses: 5 did not provide responses to both options and one respondent who considered compulsory registration of dogs was unimportant but compulsory registration of cats was very important. The model had very good fit with all observed frequencies close to expected frequencies. P for testing the null hypothesis that observed frequencies equal expected frequencies (i.e., that the model was correct) was 0.918. Dog and cat ownership status determined respondents’ views on the importance of compulsory registration. For both dogs and cats, those owning that species were less likely to consider compulsory registration of that species important compared to those not owning that species (relative risk ratio estimates 0.3 and 0.5, respectively). In addition, cat owners were more likely than non-cat owners to consider that dogs should be registered (relative risk ratio estimate 2.8). These estimates were adjusted for all four other independent variables shown in the table, so the estimated effect for dog ownership was not confounded by cat ownership and vice versa. (Any confounding by respondent’s gender, age, and their suburb’s postcode’s socio-economic status has also been removed.) However, moderate to high proportions of dog owners (80%) and cat owners (68%) considered compulsory registration of both dogs and cats was important or very important (Table 9).

Options that would apply to both species were: compulsory licensing to own a pet; on-the-spot audits and fines of pet owners with wandering cats and dogs; the development of suburbs/areas that are completely pet free. There was considerable uncertainty about the importance of suburbs/areas that are completely pet free (18%; Table 7), but more than half of the respondents (57%; Table 7) considered development of pet-free suburbs unimportant. The majority of respondents (79%) considered compulsory licensing to own a pet important or very important. Similarly, the majority of respondents (77%) considered on-the-spot audits and fines of pet owners with wandering cats and dogs important or very important.

There was little correlation between responses to development of pet-free suburbs and both compulsory licensing to own a pet and on-the-spot audits and fines (Spearman’s correlation coefficients both 0.15), but there was a moderately close correlation between compulsory licensing to own a pet and on-the-spot audits and fines (Spearman’s correlation coefficient 0.46). From PCA of these three variables (*n* = 584 respondents; the remaining 6 respondents provide responses for only 2 of these options), the first two components accounted for 46% and 32%, respectively, of the total variance of 3; eigenvalues for the first two components were 1.4 and 1.0, respectively. The first component had moderately high positive loadings for compulsory licensing to own a pet and on-the-spot audits and fines (0.68 and 0.65, respectively), but only 0.33 for pet-free suburbs, while the second component was largely determined by responses to pet-free suburbs (loadings: pet-free suburbs 0.93; compulsory licensing to own a pet −0.12; on-the-spot audits and fines −0.35). Collectively, these results indicate that respondents considered the importance of pet-free suburbs separately from compulsory licensing to own a pet and on-the-spot audits and fines. From the factor analysis, only the first factor had a positive eigenvalue. Loadings for this factor were broadly similar to those for the first principal component (0.49 and 0.46, respectively) and for compulsory licensing to own a pet and on-the-spot audits and fines (0.68 and 0.65, respectively), and only 0.19 for pet-free suburbs. Uniqueness was high for all 3 options (0.76 to 0.96), indicating the factor did not explain the variance well for any of the three options. PCA and factor analysis were repeated within females who owned dogs and/or cats (*n* = 369). The results were very similar to those described above when all 584 eligible respondents were used (results not detailed).

Based on these findings, a four-category outcome (i.e., dependent) variable was generated to capture both components. Each respondent was classified as either considering both compulsory licensing to own a pet and on-the-spot audits and fines as important or very important or not, and as considering pet-free suburbs as important or very important or not. Distributions of this response (i.e., dependent) variable were compared by five independent variables: respondent’s gender, age, dog and cat ownership status, and their suburb’s socioeconomic status. Only respondents that provided responses to all 3 options for controlling domestic (owned) dogs and cats were used for these analyses (*n* = 584).

Results of the analyses are shown in Table 10. The model had good fit with observed frequencies similar to expected frequencies for most of the 40 decile-dependent variable combinations. P for testing the null hypothesis that observed frequencies equal expected frequencies (i.e., that the model was correct) was 0.326. Females were less likely than males to consider developing pet-free suburbs important or very important (relative risk ratio estimates for females: 0.5 and 0.6). Dog owners were more likely than non-dog owners to place importance on developing pet-free suburbs but not on both compulsory licensing to own a pet and on-the-spot audits and fines of pet owners with wandering cats and dogs (relative risk ratio estimate 2.0). Respondents from higher socio-economic status postcodes (deciles 8 to 10) were more likely to place importance on either compulsory licensing to own a pet and on-the-spot audits and fines of pet owners with wandering cats and dogs (relative risk ratio estimate 4.2), and on developing pet-free suburbs (relative risk ratio estimate 2.0), but the same respondent was not more likely to select both (relative risk ratio estimate 1.1).

## 4. Discussion

In Australia, wandering dogs and cats are one of the key threats to wildlife [2,21,54,55,56,57]), which our participants were well aware of. They manifested concern and awareness towards the ‘risk of extinction of wildlife’ and, similarly, to the problem of predation by cats and dogs on native fauna [58]. This may have been because the RSPCA Qld had implemented a Living with Wildlife public education campaign prior to the current study. This campaign included information about desexing cats, confining cats and dogs, and avoiding inhumane pest control measures, and may have led to the high acceptability of different management strategies aimed at mitigating encounters between dogs/cats and wildlife. However, this campaign was only targeted at South East Queensland and would have had a limited effect throughout the state. The results about acceptability of strategies for reducing or preventing predation collectively suggest that there is a common concern that any control method is humane, but there are also widely divergent views of shooting and trapping followed by desexing and returning to the wild. Given that respondents were self-selected from the original networks of researchers, and that dog and cat owners were over-represented compared with the Queensland population, it is possible that people with relatively greater concern about animal welfare values were over-represented in our study population. The general public was targeted, however, and equally, people with strong views about cat and dog control may also have been more likely to respond to demonstrate their point of view.

Results from PCA (Table 5) identified that our respondents collectively grouped the various management strategies for reducing or preventing wild dog and cat predation on wildlife into two categories, and we interpreted these categories as strategies that cause dog and cat deaths directly due to human intervention and strategies that allow wild dogs and cats an opportunity to live a ‘natural’ life (which includes being preyed upon). This division was not a clear-cut difference between lethal and non-lethal methods because respondents collectively grouped ‘introducing a new predator’ (a lethal method) with two strategies that allow dogs and cats an opportunity to live a ‘natural’ life. This distinction needs further exploration. Zinn et al. [59] found that social norms (or shared beliefs or positions) differed around the key value-orientation towards wildlife protection or its use, which they likened to biocentric-anthropocentric values. The value orientations we found in this research may also be aligned with a biocentric-anthropocentric divide because a new predator may be seen as part of natural law (preying on species and being preyed upon), whereas other lethal strategies are directly due to human intervention. Our findings also align with Wald et al. [60] who discussed the different belief patterns between wildlife advocates who criticise non-lethal control methods and animal welfare advocates who support non-lethal control. Wald and Jacobson [61] found differences between environmentalists who felt cats were exotic animals with a high risk of damage, and TNR supporters who saw benefits from cats such as rat and mice control, and effects on humans. Further, Loyd and Miller [62] discussed groups with concerns over the effectiveness and welfare of trap and euthanise methods, and groups with concerns over TNR (favouring lethal control), stating the former hold attitudes that cats are part of nature (or have the right to hunt and reproduce because they exist in nature), and the latter hold attitudes around preserving native wildlife and quality of life.

In addition, further work is warranted on this question as this understanding is important for development and implementation of management strategies for reducing or preventing wild dog and cat predation on wildlife, and reducing potential conflicts between different normative groups [33,34]. The first two principal components in our model explained only just over half of the total variance in the seven variables used in that analysis.

Importantly, respondents’ degrees of acceptance in these categories varied markedly, showing Shackleton and colleagues’ [34] point that there is variability within normative groups as well as between. The most common responses were low acceptance for both categories (32% of respondents) and low acceptance for the first but high acceptance of the second (42%; Table 6). However, there was also a considerable percentage of respondents (18%) with high acceptance for the first but low acceptance of the second. These attitudinal differences reflect diverse value-based positions about human control of non-human animals and concern for their welfare and natural behaviours [35].

Respondents who were female, younger, and/or a dog/cat owner showed less acceptance of strategies that cause wild dog and cat deaths directly due to human intervention. These findings support the work of others who argue the existence of the strong bond between humans, dogs and cats [2,14], the prominence of animal welfare concerns [35], and social and cultural factors affect people’s opinions and attitudes about different management strategies [30,36,37]. Females and dog and cat owners were overrepresented so our descriptive results (i.e., results in Section 3.2 and Section 3.3, and Table 3) could have been affected by selection bias (i.e., could be biased relative to the entire Queensland population). This could have occurred if opinions and views differed by gender, dog ownership and/or cat ownership. However, the overrepresentation of females and dog and cat owners does not necessarily mean our relative risk ratio estimates were affected by selection bias. Such bias would depend on patterns in any overrepresentation of respondents by views about the acceptability and importance of various strategies. There is no practical way to assess these patterns within the current study. All relative risk ratios in Table 6, Table 9 and Table 10 were adjusted for all four other independent variables shown in the table, so this will have removed any confounding due to gender, dog ownership, cat ownership, age, and suburb socioeconomic status. Further research could tease out a better understanding of the differences in views about strategies that directly cause wild dog and cat deaths and those that allow wild dogs and cats to live a ‘natural’ life, as well as exploring other factors, such as whether respondents had previous experience of domestic animals through prior ownership, or working in agricultural, therapy, veterinary, pet-keeping, and related animal industries.

There is also public uncertainty about the degree of impact of dogs and cats on local wildlife (e.g., [58]). Our work aligns with others (e.g., [14,46,60]) who found gender differences in attitudes, suggesting that societal structures, such as constructions of gender (females should be more caring etc.), continue to influence the acceptability of pest control measures. Similarly, Loyd and Miller [18] found that men were more likely to support lethal methods of control, describing their beliefs as tending to be more utilitarian and anthropocentric. Previous studies have demonstrated stark differences between male and female attitudes to animals, with women showing much stronger concern for animal welfare [42,63], empathy [64], and a more positive attitude towards animals in general [65,66]. The manifestation of these gendered differences is even more evident when it comes to more gender-equitable societies [64,67], as might be the case in the Australian context.

In addition to gender, other research has shown that key predictor variables of attitudes towards animals are age and previous experiences with pets [28,45]. For example, researchers found that younger people will be less accepting of interventions that could cause harm or death to cats and dogs [68], and pet owners are less likely to manifest their support towards strategies that kill dogs and cats as the result of direct human intervention through management actions [14,45]. Age-related differences reflect the changing social values of communities at different points in time, e.g., pets are now considered part of a family, whereas previous generations may have left them outside or seen them as working animals; in contrast, different age groups have more or less time to learn about the impacts of cats and dogs on wildlife. Such changing values may also be affecting beliefs about wild dogs and cats because value orientations around both wildlife and concern for their welfare are changing [18]. In Australia, the dominant cultural understanding of wildlife as nonhuman animal species that existed prior to European colonisation is also changing to accommodate notions that cats and dogs may be also considered part of (nonhuman) nature.

Regulatory mechanisms for controlling domestic (owned) dogs and cats were considered important by most participants, and our results indicated that compulsory licensing to own a pet and on-the-spot audits and fines reflected the same latent variable. This suggests that people have a general value position around regulating pet ownership. These values were common even amongst pet-owners, which is important for management consideration. For both dogs and cats, those owning each species were less likely to consider compulsory registration of that species important compared with those not owning that species; however, moderate to high proportions of dog owners (80%) and cat owners (68%) considered compulsory registration of both dogs and cats was important or very important (Table 9). Creating expectations and norms about responsible pet ownership through on-the-spot fines or compulsory licensing to own a pet, or confining dogs and cats within privately owned spaces, are strategies that have been proven to be effective and supported by different groups of people [69].

For both dogs and cats, confinement was considered important or very important by most respondents, in contrast to Wald et al. [60] who found differences between those who believed in cat confinement and those who believed in their right to hunt. In agreement, confining dogs and cats to privately owned spaces is argued by Baird et al. (2001) (in [70]) as the best option for both wildlife protection and domestic cat welfare, with McCarthy (in [70]) suggesting community education about the benefits of domestic animal confinement. Our results indicated that respondents collectively considered the importance of pet-free suburbs separately from compulsory licensing to own a pet and on-the-spot audits and fines. While all three are regulatory options, the lack of correlation may have been due to respondents’ unfamiliarity with pet-free suburbs. This may be why, of all options offered, this option had the highest percentage of undecided respondents (18%). Although confinement of dogs and cats to privately owned spaces was highly supported, community-scale spatial solutions, such as pet-free suburbs, were considered an unimportant management option by 57% of participants. This option is, at first glance, a non-lethal action, and previous studies have reported this as a successful precautionary measure to manage owned cats/dogs wildlife encounters (Buttriss 2001 in [71]). Its lack of support warrants further investigation. Metsers and colleagues [70] noted that exclusion zones have the potential to solve, rather than simply reduce, cat populations’ impacts on wildlife, but they would need to be large enough to account for cat territorial movements and have cat-proof fencing, adequate vegetation, or other barriers. Similar to our results, Grayson et al. [72] found that cat-exclusion zones were not well supported amongst their Australian participants in new developments, and by non-cat owners, and suggested that such zones might be thought of as reducing civil liberties. There has been some success in establishing selected protected areas (Moore 2001, in [70]). In some instances, cats were observed transgressing into the cat-free space but elsewhere seen as easy to enforce (Buttriss 2001, in [70]). Older males, people of higher socio-economic status, and non-dog and non-cat owners were most likely to agree with pet-free suburbs. Lilith et al. [58] (p. 178) found similar results in responses to the statement “local governments should have the power to establish cat free zones in new sub-divisions” and noted some concerns over compliance and enforcement of several management strategies.

Some might consider agreeing to the ‘eradication’ of domestic pets from shared (human and nonhuman) urban spaces as equating, in a visual sense, to pet-free suburbs, because both are perceived as involving human interventions to kill particular animals. Females, more generally, and male pet owners, are more likely to see a continuum between domesticated and non-domesticated species, and may prefer other options that do not involve animal death in recognition of the problematic terms that ‘feral’ or ‘pest’ present and of euthanising a sentient creature [23,24,38,39]. Lethal control, particularly of animals whose presence is due to human actions, may be considered morally unacceptable by some. A distaste for, or lack of awareness of, pet-free suburbs, may reflect values that reject anthropocentric control of nonhuman animals in general.

Our research shows that respondents had a high degree of concern about human encroachment and development on wildlife habitat and were aware that human actions such as habitat fragmentation, pollution, and introduction of invasive species have a high impact on wildlife. While slightly more respondents were undecided about the impact of global warming (8%) or said it was having no impact (9%), this may have been because some respondents felt that impacts from global warming are less directly caused by humans, and that some degree of species adaptation will be possible. In addition, the question related to current impact, and it is possible that some of those who selected ‘no’ consider that there will be impacts in the future. Respondents’ lack of acceptance of strategies that cause dog and cat deaths directly due to human intervention may be influenced by the knowledge that human activities are perhaps the greatest threat to wildlife and so want to avoid ‘blaming’ companion species, and by the concept that humans should at least allow wild dogs and cats to live a ‘natural’ life given that humans introduced these species to the wildlife habitat. This may be associated with the point that we have entered into the sixth mass extinction caused by humans [73,74], and the need to tailor management actions to the underpinning socio-economic values of places with place-based solutions [58,75].

Our results suggest that, given the breadth of perspectives, there are opportunities to strategically design management actions for either reducing or preventing wild dog and cat predation on wildlife and controlling domesticated (owned) dogs and cats tailored to more specific contexts [61]. The fact that the differences between gender and age groups in attitudes towards dog and cat control are determined by dominant societal and cultural values may mean that managers could better focus their attention on designing strategies tailored to specific localities where cultural and place-based factors tend to condition gender, age, and pet-ownership norms and thus attitudes and regulations [75]. Considering the uncertainties of the different environmental variables (e.g., the predominant wildlife species being protected and social and ecological characteristics of the site), practical factors (e.g., social, cultural, and financial factors and constraints), and the receptivity of local communities to various management strategies and options, designing strategies tailored to specific localities could have greater and more sustained success than generic strategies applied across all localities.

## 5. Conclusions

While participants in this survey were highly concerned about the risks that both wild and domestic dogs and cats pose to wildlife, there was also substantial variation on the most acceptable ways to manage those that are no longer owned by a human. In particular, women and people from younger age groups were less likely to accept strategies that cause dog and cat deaths directly due to human intervention, which might reflect their more diverse views about the need for considering the welfare of all animals, structured by societal expectations. The majority of respondents considered that it is important to ensure responsible pet ownership through regulatory measures, and substantial or high proportions considered that humans were a high risk to wildlife, again suggesting some ethical underpinning concern for the animals affected by humans. These important variations in beliefs and attitudes require careful management within each community for the success of any program to manage wild dogs or cats.

## Figures and Tables

**Table 1 animals-10-01637-t001:** Distributions of demographic variables for 590 study respondents used for analyses, and for Queensland’s population in 2016.

Variable and Category	% ^1^ (Number)	% for Queensland Population in 2016
Respondent’s gender		
Male	23% (134)	51% ^2^
Female	77% (453)	49%
Other	0% (2)	
Not provided	1	
Respondent’s age (years)		
21 and under	4% (24)	7% ^2^
22 to 34	27% (160)	24%
35 to 44	21% (123)	17%
45 to 54	21% (122)	17%
55 to 64	17% (101)	15%
65 and over	10% (58)	19%
Not provided	2	
Respondent’s dog ownership		
Yes	65% (379)	37% ^3^
No	35% (207)	63%
Not provided	4	
Respondent’s cat ownership		
Yes	38% (225)	26% ^3^
No	62% (363)	74%
Not provided	2	
Respondent’s dog and cat ownership combined		
Owned both dog(s) and cat(s)	25% (144)	13% ^4^
Owned dog(s) but no cat(s)	38% (244)	25%
Owned cat(s) but no dog(s)	14% (79)	16%
Owned neither	22% (127)	46%
Response not provided for either dog or cat ownership	6	
Respondent’s suburb’s postcode’s socio-economic status ^5^		
1	6% (35)	7% ^6^
2	8% (45)	11%
3	6% (36)	8%
4	4% (25)	8%
5	13% (78)	10%
6	13% (74)	15%
7	9% (50)	11%
8	11% (63)	14%
9	15% (90)	9%
10	15% (89)	7%
Index value not available for postcode for two Queensland postcodes specified	5	

^1^ Percentages of respondents that provided a response for dog and cat ownership combined do not sum to 100% due to rounding; ^2^ Source for gender and age distributions for Queensland: Australian Bureau of Statistics (https://www.abs.gov.au/), June 2016, people aged 18 years and over only; ^3^ Source: Pet Ownership in Australia 2016, Animal Medicines Australia (https://animalmedicinesaustralia.org.au); estimated percentages of Queensland households (rather than people aged 18 years and over) from a sample of 411 people surveyed in April 2016, weighted by household location using Australian Bureau of Statistics 2016 census data; ^4^ Calculated from results in the same report as above but using statistics for households in Australia (rather than Queensland; sample size 2022 people); ^5^ Postcode’s decile based on its index of relative socio-economic advantage and disadvantage; this index aims to describe people’s access to materials and social resources, and the ability to participate in society for each postcode area in Australia; deciles are for 2630 Australian postcodes using index results from 2016; a high decile indicates relatively high advantage and relatively low disadvantage; source Australian Bureau of Statistics (https://www.abs.gov.au/); ^6^ Percentages show distribution of Queensland population (all ages) in 2016 by decile.

**Table 2 animals-10-01637-t002:** Distributions (percentages ^1^ and numbers) of 590 respondents by responses to the question “If you believe there are risks to our wildlife, what is your opinion of the impact of the following risks”?

Possible Sources of Impact	Great	Some	I Am Undecided	No	No Response Provided
Bushfires, floods, and droughts	37% (217)	59% (344)	1% (4)	4% (22)	3
Human encroachment and development in native habitats	90% (531)	9% (54)	0% (2)	1% (3)	0
Predation by cats and dogs (both wild and domestic)	71% (417)	27% (160)	1% (6)	1% (7)	0
Motor vehicle collisions	42% (249)	52% (308)	1% (4)	5% (27)	2
Global warming	34% (198)	49% (288)	8% (45)	9% (52)	7
Diseases including introduced diseases	43% (251)	53% (311)	2% (12)	2% (14)	2
Pollution	42% (245)	52% (307)	2% (12)	4% (21)	5

^1^ Percentages of respondents that provided a response for bushfires, floods, and droughts do not sum to 100% due to rounding.

**Table 3 animals-10-01637-t003:** Distributions (percentages ^1^ and numbers) of 590 respondents by views about the acceptability of various strategies for reducing or preventing wild dog and cat predation on wildlife.

Strategy to Control Dogs and Cats.	Strongly Unacceptable	Unacceptable	I Am Undecided	Acceptable	Strongly Acceptable	No Response Provided
Strategies that directly cause dog and cat deaths due to human intervention						
Shooting	23% (128)	20% (110)	6% (33)	26% (143)	25% (136)	40
Trapping followed by humane killing	10% (56)	8% (46)	4% (23)	35% (204)	44% (259)	2
Poisoning	42% (243)	30% (174)	4% (24)	13% (75)	11% (63)	11
Introducing lethal disease	44% (254)	30% (177)	8% (44)	11% (62)	8% (46)	7
Strategies that allow the dog or cat to live a ‘natural’ life						
Using reproductive control (such as immunocontraception)	5% (30)	6% (35)	5% (30)	38% (223)	46% (270)	2
Trapping followed by desexing and returning to the wild	18% (105)	18% (108)	8% (47)	29% (173)	26% (155)	2
Introducing a new predator	54% (315)	33% (191)	6% (36)	5% (30)	2% (13)	5

^1^ Percentages of respondents that provided a response for some variables do not sum to 100% due to rounding.

**Table 4 animals-10-01637-t004:** Spearman’s correlation coefficients for correlations between respondents’ views about the acceptability of various strategies for reducing or preventing wild dog and cat predation on wildlife (*n* = 541 to 588 respondents for each pair-wise comparison).

	Shooting	Trapping Followed by Humane Killing	Poisoning	Introducing Lethal Disease	Using Reproductive Control (Such as Immunocontraception)	Trapping Followed by Desexing and Returning to the Wild
Trapping followed by humane killing	0.62					
Poisoning	0.66	0.48				
Introducing lethal disease	0.58	0.43	0.72			
Using reproductive control (such as immunocontraception)	0.15	0.23	0.17	0.21		
Trapping followed by desexing and returning to the wild	−0.39	−0.31	−0.29	−0.24	0.13	
Introducing a new predator	0.18	0.11	0.27	0.36	0.10	0.10

**Table 5 animals-10-01637-t005:** Loadings for the first two principal components and the first two factors of respondents’ views about the acceptability of various strategies for reducing or preventing wild dog and cat predation on wildlife (*n* = 527 ^1^).

Strategy	Principal Component 1	Principal Component 2	Factor 1	Factor 2
Strategies that cause dog and cat deaths directly due to human intervention				
Shooting	0.49	−0.08	0.76	−0.17
Trapping followed by humane killing	0.37	−0.01	0.58	−0.15
Poisoning	0.50	0.02	0.80	0.04
Introducing lethal disease	0.47	0.10	0.71	0.18
Strategies that allow the dog or cat to live a ‘natural’ life				
Using reproductive control (such as immunocontraception)	0.10	0.57	0.20	0.23
Trapping followed by desexing and returning to the wild	−0.30	0.61	−0.23	0.44
Introducing a new predator	0.21	0.54	0.33	0.33

^1^ The remaining 63 respondents did not provide responses for all 7 strategies; 57 provided responses for 6 strategies; 6 provided responses for 5 strategies.

**Table 6 animals-10-01637-t006:** Associations between socio-demographic variables (independent variables) and respondents’ collective views about the acceptability of various strategies for reducing or preventing wild dog and cat predation on wildlife (the response or dependent variable). Each respondent was classified into one of four dependent variable categories: low or high acceptance of strategies that cause dog and cat deaths directly due to human intervention, and low or high acceptance of strategies that allow the dog or cat to live a ‘natural’ life (*n* = 527).

Acceptance of Strategies That Cause Dog and Cat Deaths Directly Due to Human Intervention:	Low	High	Low	High
Acceptance of Strategies That Allow the Dog or Cat to Live a ‘Natural’ Life:	Low	Low	High	High
% (number) of respondents	32% (170)	18% (95)	42% (219)	8% (43)
Respondent’s gender				
% ^1^ (number)				
Male	22% (27) ^2^	30% (36)	29% (35)	20% (24)
Female	35% (143)	15% (59)	45% (183)	5% (19)
Not provided			1	
Adjusted relative risk ratio (95% CI) ^3^				**<0.001** ^4^
Male		Ref. cat ^5^	Ref. cat	Ref. cat
Female		0.4 (0.2 to 0.7) 0.001	1.0 (0.6 to 1.8) 0.975	0.2 (0.1 to 0.4) <0.001
Respondent’s age (years)				
% ^1^ (number)				
34 and under	38% (63)	10% (16)	47% (78)	5% (8)
35 to 44	28% (30)	19% (20)	45% (48)	8% (8)
45 to 54	36% (41)	25% (28)	30% (34)	9% (10)
55 to 64	23% (21)	23% (21)	42% (38)	12% (11)
65 and over	29% (15)	20% (10)	39% (20)	12% (6)
Not provided			1	
Adjusted relative risk ratio (95% CI) ^3^				**0.044**
34 and under		Ref. cat	Ref. cat	Ref. cat
35 to 44		3.3 (1.4 to 7.7) 0.005	1.3 (0.7 to 2.3) 0.377	2.5 (0.8 to 8.0) 0.108
45 to 54		2.9 (1.3 to 6.2) 0.009	0.7 (0.4 to 1.2) 0.190	2.0 (0.7 to 6.0) 0.190
55 to 64		3.2 (1.3 to 7.5) 0.009	1.5 (0.8 to 2.8) 0.249	3.0 (1.0 to 9.0) 0.045
65 and over		2.1 (0.8 to 5.9) 0.149	1.0 (0.5 to 2.2) 0.957	2.0 (0.5 to 7.4) 0.309
Respondent’s dog ownership				
% ^1^ (number)				
No	27% (50)	22% (42)	41% (77)	10% (19)
Yes	36% (120)	16% (53)	42% (140)	7% (23)
Not provided			2	1
Adjusted relative risk ratio (95% CI) ^3^				**0.220**
No		Ref. cat	Ref. cat	Ref. cat
Yes		0.6 (0.3 to 1.0) 0.052	0.8 (0.5 to 1.2) 0.272	0.6 (0.3 to 1.2) 0.160
Respondent’s cat ownership				
% ^1^ (number)				
No	29% (94)	23% (73)	37% (117)	11% (36)
Yes	37% (76)	10% (21)	49% (101)	3% (7)
Not provided		1	1	
Adjusted relative risk ratio (95% CI) ^3^				**<0.001**
No		Ref. cat	Ref. cat	Ref. cat
Yes		0.4 (0.2 to 0.7) 0.002	1.0 (0.7 to 1.6) 0.920	0.2 (0.1 to 0.6) 0.003
Respondent’s suburb’s postcode’s socio-economic status ^6^				
% ^1^ (number)				
1 to 4	32% (41)	24% (30)	35% (44)	9% (12)
5 to 7	33% (58)	13% (23)	46% (82)	8% (15)
8 to 10	32% (70)	18% (39)	42% (92)	7% (16)
Index value not available for postcode or two Queensland postcodes specified	1	3	1	
Adjusted relative risk ratio (95% CI) ^3^				**0.412**
1 to 4		Ref. cat	Ref. cat	Ref. cat
5 to 7		0.6 (0.3 to 1.2) 0.150	1.4 (0.8 to 2.4) 0.272	0.9 (0.4 to 2.4) 0.913
8 to 10		0.8 (0.4 to 1.6) 0.524	1.2 (0.7 to 2.1) 0.438	0.8 (0.3 to 1.9) 0.603

^1^ Percentages of respondents in row; for some variables, percentages do not sum to 100% due to rounding; ^2^ For example, of the 122 males included, 27 (22%) considered that both sets of possible strategies had low acceptability; ^3^ Relative risk ratios were adjusted for all four other independent variables shown in the table; the model was fitted using the 515 respondents with recorded values for all five independent variables; ^4^ Bolded *p*-values are overall likelihood ratio test *p*-values for the socio-demographic variable; for example, the overall *p*-value for gender was <0.001; ^5^ Reference category; ^6^ Postcode’s decile based on its index of relative socio-economic advantage and disadvantage; this index aims to describe people’s access to materials and social resources, and the ability to participate in society for each postcode area in Australia; deciles for 2630 Australian postcodes using index results for 2016 were used; a high decile indicates relatively high advantage and relatively low disadvantage.

**Table 7 animals-10-01637-t007:** Distributions (percentages ^1^ and numbers) of 590 respondents by views about the importance of various options for controlling domestic (owned) dog and cats.

Option	Unimportant	I Am Undecided	Important	Very Important	No Response Provided
Dog-specific options					
Mandatory desexing of pet dogs	5% (27)	3% (15)	19% (110)	74% (437)	1
Confinement of dogs to the owner’s property	3% (19)	1% (6)	21% (122)	75% (443)	0
Compulsory registration of dogs by local council	11% (63)	2% (14)	21% (125)	65% (383)	5
Cat-specific options					
Mandatory desexing of pet cats	3% (16)	1% (8)	10% (57)	86% (509)	0
Confinement of cats to the owner’s property	4% (23)	2% (12)	19% (113)	75% (440)	2
Confinement of cats inside the house or in a specially constructed cat enclosure	12% (69)	6% (36)	26% (151)	57% (333)	1
Cat confinement (highest importance of the two preceding variables)	4% (23)	2% (10)	17% (101)	77% (453)	3
Compulsory registration of cats by local council	16% (92)	5% (32)	19% (114)	60% (352)	0
Cats required to wear a bell at all times	20% (119)	4% (25)	29% (170)	47% (275)	1
Options for both species					
The development of suburbs/areas that are completely pet free	57% (335)	18% (103)	15% (90)	10% (58)	4
Compulsory licensing to own a pet	16% (94)	6% (35)	27% (157)	52% (304)	0
On-the-spot audits and fines of pet owners with wandering cats and dogs	15% (91)	8% (47)	31% (180)	46% (270)	2

^1^ Percentages of respondents that provided a response: for some variables, percentages do not sum to 100% due to rounding.

**Table 8 animals-10-01637-t008:** Numbers of the 590 respondents by views about the importance of compulsory registration of dogs and cats by local councils for controlling domestic (owned) dog and cats.

Compulsory Registration of Dogs	Compulsory Registration of Cats			
	Unimportant	I Am Undecided	Important	Very Important
Unimportant	61	1		1
I am undecided		14		
Important	17	6	90	12
Very important	13	10	22	338
No response provided	1	1	2	1

**Table 9 animals-10-01637-t009:** Associations between socio-demographic variables (independent variables) and respondents’ collective views about the importance of compulsory registration of dogs and cats by local councils for controlling domestic (owned) dog and cats (the response or dependent variable). Each respondent was classified into one of three dependent variable categories: important or very important for dogs (yes or no) and important or very important for cats (yes or no) (*n* = 584).

Important or Very Important for Dogs:	No	Yes	Yes
Important or Very Important for Cats:	No	No	Yes
% (number) of respondents	13% (76)	8% (46)	79% (462)
Respondent’s gender			
% ^1^ (number)			
Male	10% (13) ^2^	8% (10)	83% (110)
Female	14% (63)	8% (36)	78% (349)
Not provided			3
Adjusted relative risk ratio (95% CI) ^3^			**0.751** ^4^
Male		Ref. cat ^5^	Ref. cat
Female		0.7 (0.3 to 1.9) 0.499	0.8 (0.4 to 1.6) 0.513
Respondent’s age (years)			
% ^1^ (number)			
34 and under	12% (21)	3% (6)	85% (154)
35 to 44	16% (20)	11% (13)	73% (90)
45 to 54	16% (19)	8% (10)	76% (91)
55 to 64	11% (11)	9% (9)	80% (81)
65 and over	9% (5)	11% (6)	81% (46)
Not provided		2	
Adjusted relative risk ratio (95% CI) ^3^			**0.107**
34 and under		Ref. cat	Ref. cat
35 to 44		2.3 (0.7 to 7.5) 0.162	0.7 (0.3 to 1.3) 0.220
45 to 54		2.3 (0.7 to 7.8) 0.175	0.6 (0.3 to 1.2) 0.180
55 to 64		3.1 (0.8 to 11.5) 0.097	0.9 (0.4 to 2.1) 0.848
65 and over		5.0 (1.1 to 23.9) 0.042	0.9 (0.3 to 2.5) 0.785
Respondent’s dog ownership			
% ^1^ (number)			
No	10% (20)	12% (24)	78% (159)
Yes	15% (56)	6% (21)	80% (300)
Not provided		1	3
Adjusted relative risk ratio (95% CI) ^3^			**0.026**
No		Ref. cat	Ref. cat
Yes		0.3 (0.2 to 0.8) 0.01	0.8 (0.4 to 1.4) 0.383
Respondent’s cat ownership			
% ^1^ (number)			
No	10% (36)	4% (13)	86% (311)
Yes	17% (39)	15% (33)	68% (151)
Not provided	1		
Adjusted relative risk ratio (95% CI) ^3^			**<0.001**
No		Ref. cat	Ref. cat
Yes		2.8 (1.2 to 6.6) 0.016	0.5 (0.3 to 0.8) 0.005
Respondent’s suburb’s postcode’s socio-economic status ^6^			
% ^1^ (number)			
1 to 4	14% (20)	9% (12)	77% (108)
5 to 7	16% (33)	8% (16)	76% (152)
8 to 10	10% (23)	8% (18)	83% (198)
Index value not available for postcode or two Queensland postcodes specified			4
Adjusted relative risk ratio (95% CI) ^3^			**0.465**
1 to 4		Ref. cat	Ref. cat
5 to 7		0.6 (0.2 to 1.7) 0.341	0.8 (0.5 to 1.6) 0.608
8 to 10		1.0 (0.4 to 2.8) 0.943	1.4 (0.7 to 2.7) 0.320

^1^ Percentages of respondents in row: for some variables, percentages do not sum to 100% due to rounding. ^2^ For example, of the 133 males included, 13 (10%) considered that compulsory registration of both dogs and cats by local councils was unimportant. ^3^ Relative risk ratios were adjusted for all four other independent variables shown in the table; the model was fitted using the 569 respondents with recorded values for all five independent variables. ^4^ Bolded *p*-values are overall likelihood ratio test *p*-value for the socio-demographic variable; for example, the overall *p*-value for gender was 0.751. ^5^ Reference category. ^6^ Postcode’s decile based on its index of relative socio-economic advantage and disadvantage; this index aims to describe people’s access to materials and social resources, and the ability to participate in society for each postcode area in Australia; deciles for 2630 Australian postcodes using index results from 2016 were used; a high decile indicates relatively high advantage and relatively low disadvantage.

**Table 10 animals-10-01637-t010:** Associations between socio-demographic variables (independent variables) and respondents’ collective views about the importance of strategies for controlling domestic (owned) dogs and cats. Each respondent was classified into one of four dependent variable categories based on their views about the importance of (a) compulsory licensing to own a pet and on-the-spot audits and fines of pet owners with wandering cats and dogs, and (b) the development of suburbs/areas that are completely pet free (*n* = 584).

Compulsory Licensing to Own a Pet and on-the-Spot Audits and Fines of Pet Owners with Wandering Cats and Dogs:	None or Only One Important or Very Important	Both Important or Very Important	None or Only One Important or Very Important	Both Important or Very Important
The Development of Suburbs/Areas That Are Completely Pet Free:	Unimportant or Undecided	Unimportant or Undecided	Important or Very Important	Important or VERY Important
% (number) of respondents	28% (164)	6% (35)	47% (273)	19% (112)
Respondent’s gender				
% ^1^ (number)				
Male	20% (26) ^2^	5% (7)	52% (69)	23% (31)
Female	31% (137)	6% (28)	45% (202)	18% (81)
Not provided	1		2	
Adjusted relative risk ratio (95% CI) ^3^				**0.088** ^4^
Male		Ref. cat ^5^	Ref. cat	Ref. cat
Female		0.7 (0.2 to 1.9) 0.475	0.5 (0.3 to 0.9) 0.015	0.6 (0.3 to 1.1) 0.079
Respondent’s age (years)				
% ^1^ (number)				
34 and under	31% (57)	7% (12)	48% (87)	14% (26)
35 to 44	26% (32)	6% (7)	54% (67)	14% (17)
45 to 54	28% (34)	8% (10)	42% (50)	22% (26)
55 to 64	28% (28)	3% (3)	45% (45)	24% (24)
65 and over	21% (12)	5% (3)	40% (23)	33% (19)
Not provided	1		1	
Adjusted relative risk ratio (95% CI) ^3^				**0.314**
34 and under		Ref. cat	Ref. cat	Ref. cat
35 to 44		1.3 (0.5 to 3.9) 0.590	1.5 (0.8 to 2.6) 0.166	1.2 (0.6 to 2.7) 0.572
45 to 54		1.4 (0.5 to 4.0) 0.503	0.9 (0.5 to 1.6) 0.676	1.5 (0.8 to 3.1) 0.238
55 to 64		0.6 (0.1 to 2.2) 0.406	1.0 (0.5 to 1.8) 0.970	1.5 (0.7 to 3.2) 0.274
65 and over		0.9 (0.2 to 4.7) 0.887	1.1 (0.5 to 2.5) 0.762	2.6 (1.1 to 6.4) 0.033
Respondent’s dog ownership				
% ^1^ (number)				
No	32% (66)	7% (15)	37% (77)	23% (48)
Yes	26% (97)	5% (20)	52% (194)	17% (63)
Not provided	1		2	1
Adjusted relative risk ratio (95% CI) ^3^				**0.004**
No		Ref. cat	Ref. cat	Ref. cat
Yes		1.1 (0.5 to 2.4) 0.872	2.0 (1.3 to 3.1) 0.001	1.1 (0.6 to 1.8) 0.84
Respondent’s cat ownership				
% ^1^ (number)				
No	24% (87)	4% (16)	48% (174)	23% (83)
Yes	34% (76)	8% (18)	45% (99)	13% (29)
Not provided	1	1		
Adjusted relative risk ratio (95% CI) ^3^				**0.029**
No		Ref. cat	Ref. cat	Ref. cat
Yes		1.4 (0.6 to 3.1) 0.413	0.7 (0.5 to 1.1) 0.167	0.5 (0.3 to 0.9) 0.012
Respondent’s suburb’s postcode’s socio-economic status ^6^				
% ^1^ (number)				
1 to 4	33% (46)	3% (4)	40% (56)	24% (34)
5 to 7	32% (65)	4% (9)	47% (94)	16% (33)
8 to 10	22% (53)	8% (19)	51% (123)	18% (44)
Index value not available for postcode or two Queensland postcodes specified		3		1
Adjusted relative risk ratio (95% CI)^3^				**0.011**
1 to 4		Ref. cat	Ref. cat	Ref. cat
5 to 7		1.4 (0.4 to 4.8) 0.640	1.2 (0.7 to 2.0) 0.485	0.7 (0.4 to 1.4) 0.319
8 to 10		4.2 (1.3 to 13.6) 0.015	2.0 (1.2 to 3.4) 0.010	1.1 (0.6 to 2.0) 0.827

^1^ Percentages of respondents in row: for some variables, percentages do not sum to 100% due to rounding. ^2^ For example, of the 133 males included, 26 (20%) selected responses as shown in the column headings. ^3^ Relative risk ratios were adjusted for all four other independent variables shown in the table; the model was fitted using the 570 respondents with recorded values for all five independent variables. ^4^ Bolded *p*-values are overall likelihood ratio test *p*-value for the socio-demographic variable; for example, the overall *p*-value for gender was 0.088. ^5^ Reference category. ^6^ Postcode’s decile based on its index of relative socio-economic advantage and disadvantage; this index aims to describe people’s access to materials and social resources, and the ability to participate in society for each postcode area in Australia; deciles for 2630 Australian postcodes using index results from 2016 were used; a high decile indicates relatively high advantage and relatively low disadvantage.

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
