# Peer review of "Beliefs and Attitudes of Residents in Queensland, Australia, about Managing Dog and Cat Impacts on Native Wildlife"

_animals, 2020, doi:10.3390/ani10091637_

Round 1

Reviewer 1 Report

These revisions are adequate.

Author Response

Authors’ response:  Thank you very much.

Reviewer 2 Report

Beliefs and Attitudes of Residents in Queensland, Australia, about Managing Dog and Cat Impacts on Native Wildlife

Line 10: management actions for controlling dogs and cats….. could this be changed to management actions for controlling dog and cat populations and behaviour (hereafter ‘controlling dogs and cats). This would be a more accurate sentence as both the individual’s behaviour is being controlled, and the population number directly.

General comment why not just label the populations ‘feral’ ‘feral hybrid’ (domestic dogXdingo) and ‘owned’ which better reflects the situation rather than wild and domestic, and is easier for the reader to understand.

Line 27 – can the measure of acceptability be briefly explained? Is it a Likert score?  

Keywords – suggest adding population control

Line 50 – needs citation and should be specific to domestic dogs (to separate from dingo).

Line 57 ‘to European colonisation)’ – citation needed for this cultural definition.

Line 62 – add the date to the legislation title, 2014.

Line 67 delete ‘On the one hand’

Line 140 – please define TNR the first time you use it.

General comment on Introduction – extremely informative, well researched, and well written.

Line 204 – what modifications were made following the pilot?

An ethics statement is imperative – who ethically reviewed this and what permissions/consent was achieved? Were participants briefed with an information sheet?

Given the distribution methods of the survey link, and that very similar people likely answered this questionnaire, did the list of demographics considered the well-established factors that influence values/beliefs regarding animals? For example, were the following questions asked: have you ever owned a dog or cat? Were you raised with pet/companion animals (if so which ones)? Do you consider yourself to be an animal lover? Do you work in the agricultural industry? Do you work in the pest control industry? Do you work in animal advocacy? If so, I hope this is explored more in the results section, but if not this should be considered in the Discussion as there is likely some sampling bias evident in the recruited sample.

Line 228 and 231 – principal component analysis (no s on component) and can abbreviate to PCA hereafter.

You can run a PCA directly on ordinal data if you use the maximum likelihood method rather than the Pearson’s correlation method. Or you could pre-test your Likert scores for normality and see if they are normally distributed and if so run a standard PCA. Might be worth investigating. However the way the PCA has been conducted here is acceptable, it might mean some finer detail results are overlooked and this should be acknowledged in the discussion.

Overall comment on the Methods – very clearly written and good level of detail. More on ethics needed.  

Line 261 ‘own), and 53 respondents that replied to less than 29 of these questions were excluded. Of the’,  change to ‘own), and 53 respondents who replied to less than 29 of these questions were excluded. Of the’

Given the demographic results, this is essentially a survey of female dog and/or cat owners. If just data for these respondents are used do you get the same results? Is it worth running the analysis with and without male and non-dog and//or -cat owners to see if these muddy the analysis? this needs to be considered and discussed in the Discussion otherwise you are currently overgeneralising your results. The results in Table 6 (and to a lesser extent Table 9) suggest dog and cat ownership would not cause a difference in results but that male vs. female would. This needs further consideration therefore.

Line 310  ‘closely correlated’ is this the same as ‘significantly correlated’? Significantly would be easier to understand for the reader.

Line 348 why were the scores for PC1 and PC2 not used in further analysis? These scores should be correlated with the demographics and responses to other questions. This would be worth doing given the results you found that demographics have no relationship with man-made vs natural control methods. Same is true for Line 468 – why not use the PC scores?

Overall comment on Results – great presentation of data and use of tables. Very clearly written but need to think about the demographic bias and how this could be influencing the results and explain why PC scores are not used in further analysis.

Line 510 can other potential influences be mentioned? The RSPCA Qld campaign may have had an influence but the Introduction covers many other potential influences and these need briefly discussing here otherwise the potential influence of the RSPCA Qld campaign is overstated as this is not a before and after design.

In the Discussion more could be made of the results in table 2 – respondents are most concerned about human encroachment and development, yet they are less concerned with natural disaster or global warming and pollution …… maybe they are distancing themselves from causes they will be responsible for. Maybe they see that the issues they are most concerned about they can control more easily like keeping your dog on a lead or de-sexing it.

I am glad to see that both sex differences and dog/cat owner demographics have been considered however there are sample biases that have not been fully overcome or investigated – gender, dog /cat ownership – the sample does not reflect the target population and from the literature owning a pet is important in what view you have about animal treatment and control. This needs further investigation in the results (do the analysis with just the non dog/cat owners and see if you get different results OR discuss it in more detail in the Discussion. Here also, there should be consideration of other demographic variables that should have been considered – like if the respondent had other pets or had previously owned a dog/cat, or worked in the agricultural industry etc.

Overall an excellent manuscript with only a small amount of further consideration needed regarding sample biases.

Author Response

Reviewer 2:

Line 10: management actions for controlling dogs and cats….. could this be changed to management actions for controlling dog and cat populations and behaviour (hereafter ‘controlling dogs and cats). This would be a more accurate sentence as both the individual’s behaviour is being controlled, and the population number directly.

Authors’ response:   changed to “management actions for controlling dog and cat populations (hereafter managing dogs and cats) as we think managing is more appropriately and used more widely.  It includes controlling but we also critique the idea of the need to control elsewhere in the paper.

General comment why not just label the populations ‘feral’ ‘feral hybrid’ (domestic dogXdingo) and ‘owned’ which better reflects the situation rather than wild and domestic, and is easier for the reader to understand.

Authors’ response: This is an interesting point, although most legislation refers to pest animals that have not hybridised with native animals as it is an umbrella term for a range of species. 

Line 27 – can the measure of acceptability be briefly explained? Is it a Likert score?  

Authors’ response: Yes, acceptability of various strategies was assessed using 5-category Likert scores. We have added '(each assessed using 5-category Likert scores)' to line 27.  Also we have added “Community” before acceptability and acceptability was explored in the survey as explained in methodology.

Keywords – suggest adding population control

Authors’ response. The authors don’t believe this is necessary as we are more concerned about people’s beliefs about management.

Line 50 – needs citation and should be specific to domestic dogs (to separate from dingo).

Authors’ response:  The authors have added “(Jones 1983, Newsome 1983) with Newsome (1983) at that point suggesting dogs as the same species as dingos given they can interbreed.”

Line 57 ‘to European colonisation)’ – citation needed for this cultural definition.

Authors’ response:  Added “as noted by relevant legislation, but see Carter and Paterson, 2021

Line 62 – add the date to the legislation title, 2014.

 Authors’ response:  the year 2014 added to the title.

Line 67 delete ‘On the one hand’

 Authors’ response:  “On the one hand” removed.

Line 140 – please define TNR the first time you use it.

Authors’ response:  We have added changed this to: Trap-Neuter-Release, known as TNR [30,37,38] programs. This strategy is usually only used for cats and involves trapping them, neutering them and releasing them back to where they were trapped. It is favoured by people opposed to euthanasia.

General comment on Introduction – extremely informative, well researched, and well written.

 Authors’ response:  Thank you very much!

Line 204 – what modifications were made following the pilot?

 Authors’ response:  It has been changed to “Some modifications were made to a few questions to improve clarity and remove any ambiguity following this pilot testing.”

An ethics statement is imperative – who ethically reviewed this and what permissions/consent was achieved? Were participants briefed with an information sheet?

Authors’ response:   We have added “The research was carried out under the auspices of RSPCA Qld and adhered to the National Statement on Ethical Conduct of Research 2007-2015 and the update 2007-2018, and also the Australian Code for the Responsible Conduct of Research 2018.”​ at the beginning of 2.1.

The email invitation included information on the research. Participation was completely voluntary

Given the distribution methods of the survey link, and that very similar people likely answered this questionnaire, did the list of demographics considered the well-established factors that influence values/beliefs regarding animals? For example, were the following questions asked: have you ever owned a dog or cat? Were you raised with pet/companion animals (if so which ones)? Do you consider yourself to be an animal lover? Do you work in the agricultural industry? Do you work in the pest control industry? Do you work in animal advocacy? If so, I hope this is explored more in the results section, but if not this should be considered in the Discussion as there is likely some sampling bias evident in the recruited sample.

Authors’ response: We decided we wanted to keep the survey as brief as possible so only included information about their current pet ownership. We were aware of these other questions you mention but thought we would keep it simple. We thought that most animal lovers would have a current pet and therefore used current pet ownership as a proxy for the above questions.

Line 228 and 231 – principal component analysis (no s on component) and can abbreviate to PCA hereafter.

 Authors’ response: Edited. Thank you.

You can run a PCA directly on ordinal data if you use the maximum likelihood method rather than the Pearson’s correlation method. Or you could pre-test your Likert scores for normality and see if they are normally distributed and if so run a standard PCA. Might be worth investigating. However the way the PCA has been conducted here is acceptable, it might mean some finer detail results are overlooked and this should be acknowledged in the discussion.

Authors’ response: Many thanks for these suggestions. Prompted by these, we have explored and used factor analysis (although not the maximum likelihood method as we were not convinced that our data met the assumptions for this approach), and have added results from those analyses alongside results from the principal components analyses.

Overall comment on the Methods – very clearly written and good level of detail. More on ethics needed.  

 Authors’ response:  Thank you very much.

Line 261 ‘own), and 53 respondents that replied to less than 29 of these questions were excluded. Of the’,  change to ‘own), and 53 respondents who replied to less than 29 of these questions were excluded. Of the’

Authors’ response:  changed to “who”.

Given the demographic results, this is essentially a survey of female dog and/or cat owners. If just data for these respondents are used do you get the same results? Is it worth running the analysis with and without male and non-dog and//or -cat owners to see if these muddy the analysis? this needs to be considered and discussed in the Discussion otherwise you are currently overgeneralising your results. The results in Table 6 (and to a lesser extent Table 9) suggest dog and cat ownership would not cause a difference in results but that male vs. female would. This needs further consideration therefore.

Authors’ response: Another great suggestion. We have repeated both the PCA and factor analysis limited to females who owned dogs and/or cats. Results for this subset were remarkably similar to these when all eligible respondents were included. We have added text to both Statistical Methods and Results describing these additional analyses.

Line 310  ‘closely correlated’ is this the same as ‘significantly correlated’? Significantly would be easier to understand for the reader.

Authors’ response: This is a difference between evidence for presence of an association (for correlation coefficients, evidence that there is truly a correlation) and the likely magnitude of the association (for correlation coefficients, the closeness of the correlation). They can be quite different. A correlation coefficient can be significant (ie the p-value is low) thus, there is evidence that there is truly a correlation but that correlation can be weak (ie not close) and of little practical importance. The closeness of the correlation is of primary interest to us so our interest is in the magnitude of the correlation coefficient estimates.

We have added text at the first use of the ides of close correlation '(i.e. where the absolute value of the correlation coefficient estimate was closest to 1)'.

Line 348 why were the scores for PC1 and PC2 not used in further analysis? These scores should be correlated with the demographics and responses to other questions. This would be worth doing given the results you found that demographics have no relationship with man-made vs natural control methods. Same is true for Line 468 – why not use the PC scores?

Authors’ response: Yes, we initially considered this and there would certainly have been statistical advantages in this approach. However, there would also have been substantial communication challenges in communicating results of regression analyses with PC1 and PC2 as dependent variables. For example, a result could be that PC1 is estimated as being x units higher for females relative to males. Few readers would be able to work our whether x units is large or small, and even what a 1 unit difference means. For this reason, we used variables directly collected as dependent variables, rather than principal components, as dependent variables.

Overall comment on Results – great presentation of data and use of tables. Very clearly written but need to think about the demographic bias and how this could be influencing the results and explain why PC scores are not used in further analysis.

Authors’ response: Thank you and please see our comment above which should have improved the paper.

Line 510 can other potential influences be mentioned? The RSPCA Qld campaign may have had an influence but the Introduction covers many other potential influences and these need briefly discussing here otherwise the potential influence of the RSPCA Qld campaign is overstated as this is not a before and after design.

Authors’ response:  A sentence “However, this campaign was only targeted at South East Queensland and would have had limited effect throughout the state.” Was added at Line 102.

In the Discussion more could be made of the results in table 2 – respondents are most concerned about human encroachment and development, yet they are less concerned with natural disaster or global warming and pollution …… maybe they are distancing themselves from causes they will be responsible for. Maybe they see that the issues they are most concerned about they can control more easily like keeping your dog on a lead or de-sexing it.

Authors’ response:  pollution is also included in the sentence “Our research shows that respondents had a high degree of concern about human encroachment and development on wildlife habitat, and were aware that human actions such as habitat fragmentation, pollution, and introduction of invasive species have a high impact on wildlife”.  We have added the sentence  “While slightly more respondents were undecided about the impact of global warming (8%) or said it was having no impact (9%), this may have been because some respondents felt impacts from global warming are less directly caused by humans, that some degree of species adaptation will be possible. In addition, the question related to current impact, and it is possible that some of those who selected no consider that there will be impacts in the future.”

I am glad to see that both sex differences and dog/cat owner demographics have been considered however there are sample biases that have not been fully overcome or investigated – gender, dog /cat ownership – the sample does not reflect the target population and from the literature owning a pet is important in what view you have about animal treatment and control. This needs further investigation in the results (do the analysis with just the non dog/cat owners and see if you get different results OR discuss it in more detail in the Discussion.

Authors’ response: All relative risk ratios in Tables 6, 9 and 10 were adjusted for all four other independent variables shown in the table. So this will have removed any confounding due to gender, dog ownership, cat ownership, age and suburb's socioeconomic status. Females and dog and cat owners were overrepresented so descriptive results (i.e. results in sections 3.2 and 3.3, and Table 3) could have been affected by selection bias (ie could be biased relative to the entire Queensland population). This could have occurred if opinions about the impact of various risks (Table 2) and views about the acceptability of various strategies for reducing or preventing wild dog and cat predation on wildlife (Table 3) differ by gender, dog ownership and/or cat ownership.

The overrepresentation of females and dog and cat owners does not necessarily mean our relative risk ratio estimates were affected by selection bias. Such bias would depend on patterns in any overrepresentation of respondents by views about the acceptability and importance of various strategies. There is no practical way to assess these patterns with the current study.

We have added a paragraph to Discussion discussing bias:

"Females and dog and cat owners were overrepresented so our descriptive results (i.e. results in sections 3.2 and 3.3, and Table 3) could have been affected by selection bias (i.e. could be biased relative to the entire Queensland population). This could have occurred if opinions and views differ by gender, dog ownership and/or cat ownership. However, the overrepresentation of females and dog and cat owners does not necessarily mean our relative risk ratio estimates were affected by selection bias. Such bias would depend on patterns in any overrepresentation of respondents by views about the acceptability and importance of various strategies. There is no practical way to assess these patterns within the current study. All relative risk ratios in Tables 6, 9 and 10 were adjusted for all four other independent variables shown in the table, so this will have removed any confounding due to gender, dog ownership, cat ownership, age and suburb's socioeconomic status."

Here also, there should be consideration of other demographic variables that should have been considered – like if the respondent had other pets or had previously owned a dog/cat, or worked in the agricultural industry etc.

Authors’ response:  We have added “Further research could tease out better understanding of differences in views about strategies that directly cause wild dog and cat death and those that allow wild dogs and cats to live a ‘natural’ life, as well as exploring other factors such as whether respondents had previous experience of domestic animals through prior ownership or working in agricultural, therapy, veterinary, pet-keeping and related animal industries.”

Overall an excellent manuscript with only a small amount of further consideration needed regarding sample biases.

Authors’ response:  Thank you for your consideration, we hope this is an improved manuscript.

This manuscript is a resubmission of an earlier submission. The following is a list of the peer review reports and author responses from that submission.

Round 1

Reviewer 1 Report

Dear Editor

I have finished reviewing the manuscript by Carter et al. They adress and interesting an important topic of key importance for wildlife conservation in many areas of the world.

While I value the research conducted, I think that this article needs to clarify and re-organize both the methods and the results. In the current version these sections are difficult to follow. I also think that some of the data need to be re-analyzed (e.g., including multiple predictor variables in the same model, or changing the analytical approach) before publication.

I hope that my comments below help to improve the manuscript.

Introduction

In general terms, the introduction is correct. However, it needs to provide more theoretical background as well as to explicitly state objectives. Specific comments below.

L. 54, I would prefer to use "stray" over "feral" or "wild", because many of the unowned stray dogs/cats may not be feral nor wild. Complete definitions can be found in Vanak & Gompper (2009) Mammal Review and in the OIE (2019) Health Code.

L. 54-55, mechanisms should follow order of importance. Both for dogs and cats predation is the main mechanism (see Doherty et al. 2016, PNAS; Doherty et al. 2017, Biol. Conserv.). In addition, "species behavioral changes" should be listed either as "disturbance" or "non-lethal effects of predation".

L. 93-105, I think that the theoretical background of the drivers of acceptability is weak and needs to be strengthened. One of the main theoretical bodies that could be used is the Theory of Planned Behavior. Also, the Cognitive Hierarchy, is widely use to assess acceptability of (wildlife) management interventions  (check authors such as Manfredo and Vaske). There is some useful literature that I strongly recommend:

For dogs:

Miller, K. K., Ritchie, E. G., & Weston, M. A. (2014). The human dimensions of dog-wildlife interactions. Free ranging dogs and wildlife conservation, 286-301.

Williams, K. J., Weston, M. A., Henry, S., & Maguire, G. S. (2009). Birds and beaches, dogs and leashes: Dog owners' sense of obligation to leash dogs on beaches in Victoria, Australia. Human Dimensions of Wildlife14(2), 89-101.

Schneider, T. J., Maguire, G. S., Whisson, D. A., & Weston, M. A. (2020). Regulations fail to constrain dog space use in threatened species beach habitats. Journal of Environmental Planning and Management63(6), 1022-1036.

Díaz, M. V., Simonetti, J. A., & Zorondo-Rodríguez, F. (2020). Social acceptability of management actions for addressing different conflict scenarios between humans and wildlife in Patagonia. Human Dimensions of Wildlife25(1), 17-32.

For cats:

Wald, D. M., Jacobson, S. K., & Levy, J. K. (2013). Outdoor cats: Identifying differences between stakeholder beliefs, perceived impacts, risk and management. Biological Conservation167, 414-424.

Loyd, K. A. T., & Miller, C. A. (2010). Influence of demographics, experience and value orientations on preferences for lethal management of feral cats. Human Dimensions of Wildlife15(4), 262-273.

Wald, D. M., & Jacobson, S. K. (2014). A multivariate model of stakeholder preference for lethal cat management. PloS one9(4), e93118.

General:

Crowley, S. L., Hinchliffe, S., & McDonald, R. A. (2017). Conflict in invasive species management. Frontiers in Ecology and the Environment15(3), 133-141.

Engel, M. T., Vaske, J. J., Bath, A. J., & Marchini, S. (2017). Attitudes toward jaguars and pumas and the acceptability of killing big cats in the Brazilian Atlantic Forest: An application of the Potential for Conflict Index 2. Ambio46(5), 604-612.

Estévez, R. A., Anderson, C. B., Pizarro, J. C., & Burgman, M. A. (2015). Clarifying values, risk perceptions, and attitudes to resolve or avoid social conflicts in invasive species management. Conservation Biology29(1), 19-30.

Shackleton, R. T., Richardson, D. M., Shackleton, C. M., Bennett, B., Crowley, S. L., Dehnen-Schmutz, K., ... & Marchante, E. (2019). Explaining people's perceptions of invasive alien species: a conceptual framework. Journal of Environmental Management229, 10-26.

Whittaker, D., Vaske, J. J., & Manfredo, M. J. (2006). Specificity and the cognitive hierarchy: Value orientations and the acceptability of urban wildlife management actions. Society and Natural Resources19(6), 515-530.

Zinn, H. C., Manfredo, M. J., Vaske, J. J., & Wittmann, K. (1998). Using normative beliefs to determine the acceptability of wildlife management actions. Society & Natural Resources11(7), 649-662.

L. 119-121, check grammar.

L- 140-145. I suggest to explicitly state the objectives of the study. Currently the manuscript states "we explore socio-demographic factors that may shape human attitudes towards dog and cat management strategies". I suggest to show some a priori thinking on which factors are going to be explored.

Methods

Methods need more detail. References for methods used are in most cases not provided. This section needs major improvement.

In the first section authors do not provide details regarding how they designed the questionnaire. For example, there are no references to previous research that could have been useful to design the instrument. Furthermore, there is no detail regarding the type of question. DId they use Likert-type scales? Or open questions? Or multiple choice? This information is very important. Adding the questionnaire as supplementary material would also be helpful.

The section on participant enrollment also needs more detail. Authors state that "participants were recruited from the general public rather than from known animal lovers and followers of RSPCA Qld". They used snowball sampling, but it appears that stakeholders selected were contacts of the second author that is affiliated to the Royal Society for the Prevention of Cruelty to Animals. This appears as a contradiction to the statement that the public was not recruited from followers of RSPCA. This statement suggest that selected participants could be in fact linked to RSPCA. Therefore, I am concerned about the potential for (1) sampling bias, and (2) desirability bias. Even if participants were not RSPCA followers, the fact that the interview is linked to RSPCA could lead to response biases. In  any case, potential biases need to be discussed, especially considering that the participants are not a representative sampling from the population as suggested by Table 1.

Data analysis: This section also needs clarification. First, here we have the first glimpse to the structure of the data collected, but it is hard  to judge the methods without complete information on data collected. Second, given that the objectives are not entirely clear, it is also unclear the decision to use PCA versus other analyses. If the purpose is to assess internal validity of the scales, I suggest to use Cronbach alpha, rather than PCA. Regarding the decision to transform ordinal data into binary responses, this needs further justification. For example, why are the unacceptable responses colapsed with undecided? There may be justification for this, but needs to be provided. 

For the multinomial logistic regression it is unclear what are the response variables. Furthermore, no information is provided regarding model selection (if conducted), assessment of potential multicollinearity, etc. Authors report univariate analyses. In my opinion, it is important to conduct the analyses including all variables that a priori could be important. Authors can decide to do model selection (hopefully through information theoretic approaches) or to present the full models. Both methods and results for this analysis need much more clarification.

From the results (this is not clear in methods, but it should be), authors combine responses to two different types of question to create 4 categories (e.g., low-low, low-high; high-low; high-high in Table 6, see also Tables 9 and 10). Authors could instead use mixed models, therefore facilitating the interpretation of results.   

Finally, references are mostly missing as is the case for most of the methods section.

Results

Results need more clarity. The fact that neither methods nor objectives are clear makes difficult to follow this section. In addition, there are sections that correspond to methods inside the results sections. Tables are very large and it is unclear what are the statistical analyses they are reporting. In the case of tables, I suggest expanding it horizontally to reduce the number of rows. It would be easier to interpret RR and p values if they were presented in the same row than the variables they are evaluating as traditionally done in most articles.

The first paragraph in section 3.1. could be moved to Methods (there are other sections that also report methods) 

PCA: Eigenvalues not reported, this is important because we would like to know the criteria used for deciding how many components would be retained. Note that the two components extracted explain only about half of the variance. Therefore, I would be more cautious regarding the interpretation of the components (loadings are not bad, but not very high either).

Multinomial logistic regression: No measures of model fit are presented. The presentation of tables makes difficult to follow the results. As stated in the methods section, authors need to include the different predictor variables simultaneously in the models, and hopefully conduct model selection. I would consider using mixed effects models instead of creating categories through the combination of answers to different questions. This would make easier to understand drivers of lethal vs non lethal management alternatives (or the other combinations presented).

L. 283-291, correspond to methods. I think that authors should use Cronbach alpha, to assess item validity, before this analysis. I think it is more adequate than PCA for these objectives.

L. 260-265, corresponds to methods.

Table 6. It is unclear the analysis behind this table (same for tables 9 and 10). The RR CI for females in the combination high-high seems to have error (the mean equals the lower bound of the CI). 

Discussion

Like the introduction, the discussion is generally correct. My main concerns are:

  1. The problems outlined in methods and results, makes difficult to judge the soundness of the discussion. Improving the analysis, as well as the reporting of methods and results, will help to clarify this.
  2. Authors acknowledge sampling bias, and this is good. I think they should also acknowledge and discuss potential desirability bias.
  3. The manuscript would benefit from a discussion that includes more theoretical background, including the broader understanding of drivers of wildlife management acceptability. The references provided above (and certainly other references) may be useful. 

Reviewer 2 Report

This study surveys nearly 600 people, both cat/dog owners and non-owners, to explore attitudes about the acceptability of different strategies to control feral and domestic cats and dogs, and acceptance of strategies that either result in cat/dog deaths or that allow the cat/dog to live a natural life. They also explore the extent to which socio-demographic variables influence attitudes towards some control measures. The data have been robustly analysed, and the manuscript is well-written, if rather dense.

I felt that the summary and abstract did not adequately represent the content of the paper. The 2nd last sentence in the simple summary was very hard to understand without having read the rest of the paper. The same goes for the final sentence of the first paragraph of the abstract: why was it expected that participants would group their responses one way as opposed to the other?

In the Discussion (first paragraph) it would be useful to know something about regulations/public education initiatives in the area sampled. Concern about extinction risk for wildlife and acceptability of control measures seems very high: is this because of programmes that have been in place for several years educating people about impacts of cats on wildlife?

The authors do acknowledge the biases in their data, both with respect to whether respondents owned dogs/cats, and also with respect to gender. These biases might explain some of the results with respect to acceptance of control methods. Similar studies have been done in NZ on acceptability of different control strategies for the invasive brushtail possum. Maybe the authors could consider other studies in their assessment of the impact of their biases on their findings.

I found the large tables hard to interpret at first, but once I had one figured out they seemed to be a logical way to present these data.

Minor points

Line 52: It is not clear how relationships with cats and dogs all ow the development of more respectful relationships with the environment. Could the authors provide a little more detail?

Line 66. “widely” not ‘wildly”

Line 71. “lives” not “life”.

Line 72. Why is “human killing” in inverted commas?

Line 76. Could the authors expand a little on these uncertainties with respect to relationship with environmental factors. Perhaps they could provide an example.

Line 90. Can the authors provide some citations to back up their statement about “contentious and debated topic”?

Line 100. Delete “about”.

Line 101-102. Perhaps rewrite as “There are public concerns that limiting the spatial ranges of dogs and cats effectively restricts some natural behaviours”.

Line 103. “the welfare of native wildlife”.

Line 104. Not clear what is meant by “between these positions”.

Line 114. “domestic animals which” not “who”.

Line 128. Not clear why there is “however” as it doesn’t seem to be contrasting.

Line 148. Where in Queensland? Be more specific.

Line 200. Why was it so important that respondents were from Queensland that the authors were willing to exclude 151 responses? Little context is provided about how people in Queensland’s attitudes might be shaped by initiatives or local regulations around pet ownership.

Line 208. 3.2 Risk of extinction.

Line 231. Stipulate in heading whether this section is about feral or domestic/owned animals.

Line 243. Delete “?” after “percentages”.

Line 316. Should be 3.5.

Line 378. “component”, not “components”

Line 420.

Line 452. Delete “that”.

Line 456-458. Maybe older people have had more time to learn about the negative impacts of cats and dogs on native wildlife.

Author Response

Please see attached report.

Reviewer 3 Report

Authors provide insight to age and gender differences in attitudes toward feral dogs and cats in Queensland area.    When talking about 'wild life' in relation to feral dogs or cats in Queensland, the image of it may differ very much from the rest or the world.  The amount of feral dogs and cats may also different.  Would it be possible to mention what wild life animal here means?  The land use is also different from other Western nations that the image of feral dogs or cats in the wild may be different from the northern hemisphere nations.  Some further mention of these things may help image the situation in Queensland.  In the conclusion section, authors talks about the cultural differences.  I believe writing about these environmental differences are needed in order to emphasize the need of controlling the feral dogs and cats.
